# Additive Manufacturing Processes in Medical Applications

**DOI:** 10.3390/ma14010191

**Published:** 2021-01-03

**Authors:** Mika Salmi

**Affiliations:** Department of Mechanical Engineering, Aalto University, 02150 Espoo, Finland; mika.salmi@aalto.fi

**Keywords:** additive manufacturing, rapid manufacturing, rapid prototyping, 3D printing, medical, implants, dental, processes, methods, clinical

## Abstract

Additive manufacturing (AM, 3D printing) is used in many fields and different industries. In the medical and dental field, every patient is unique and, therefore, AM has significant potential in personalized and customized solutions. This review explores what additive manufacturing processes and materials are utilized in medical and dental applications, especially focusing on processes that are less commonly used. The processes are categorized in ISO/ASTM process classes: powder bed fusion, material extrusion, VAT photopolymerization, material jetting, binder jetting, sheet lamination and directed energy deposition combined with classification of medical applications of AM. Based on the findings, it seems that directed energy deposition is utilized rarely only in implants and sheet lamination rarely for medical models or phantoms. Powder bed fusion, material extrusion and VAT photopolymerization are utilized in all categories. Material jetting is not used for implants and biomanufacturing, and binder jetting is not utilized for tools, instruments and parts for medical devices. The most common materials are thermoplastics, photopolymers and metals such as titanium alloys. If standard terminology of AM would be followed, this would allow a more systematic review of the utilization of different AM processes. Current development in binder jetting would allow more possibilities in the future.

## 1. Introduction

Additive manufacturing (AM), or, in a non-technical context, 3D printing, is a process where physical parts are manufactured using computer-aided design and objects are built on a layer-by-layer basis [1]. Usually, these procedures are called toolless processes. There are other processes, such as incremental sheet forming or laser forming, that build objects on a layer-by-layer basis as well but do so by adding the form, not the material [2,3]. These processes are not counted as an additive manufacturing process even though they have been similarly used in making, for example, customized medical products [4,5]. Currently, additive manufacturing is utilized and being investigated for use in areas such as the medical, automotive, aerospace and marine industries, as well as industrial spare parts [6,7,8,9,10]. Additive manufacturing is referred to as a manufacturing method where complexity or customization is free [11]. However, this requires marking and tracing of the different parts compared to mass production of the same kind of parts. Nevertheless, when comparing AM against conventional manufacturing, it has a much higher potential for customization and complex geometries. However, when comparing cost, additive manufacturing is usually not cheaper if the geometry is designed for mass production and only the manufacturing cost is calculated [12]. It would suffice to reiterate the whole product design and look at the economics over the entire product lifecycle [13]. AM is currently developing fast, and new players are entering the market all the time. There have been substantial investments in new companies, such as Carbon, Desktop Metal and Formlabs, as well as internal development in large companies in other areas, such as HP and GE. Even though the basic principles of the different AM processes have stayed the same, there are now more development resources to take the next step forward for these technologies, and this will also open up new possibilities in medical applications [14,15,16,17].

In the medical field, every patient is unique, and therefore, AM has a high potential to be utilized for personalized and customized medical applications. The most common medical clinical uses are personalized implants, medical models and saw guides [18]. In the dental field, AM is utilized on splints, orthodontic appliances, dental models and drill guides. However, AM has also been explored for making artificial tissues and organs [19]. In medicine, there is a background in digitalization of medical imaging, and that digitalization allows for reconstructing 3D models from patients’ anatomy. A typical workflow for personalized medical devices starts with imaging or capturing the patient’s geometry using computed tomography or other 3D scanning methods [20]. Then, these data are manipulated to obtain a 3D model of the patient’s anatomy, and this can be an example already of additive manufacturing such as a medical model. Moreover, the geometry can be utilized to design patient-specific implants, and this design can be additively manufactured. After manufacturing, there is quite often a need for post-processing, such as polishing [21]. When the medical device is ready, the final step is the clinical application and follow-up. 

The usage of AM is usually related to the question of what the benefits are compared to existing processes and technologies. Most often, the questions are related to whether it is cheaper to manufacture, but the whole lifecycle of the product and process should be investigated. The actual manufacturing prices cannot be the only performance indicator. Table 1 summarizes some of the benefits of AM in the medical and dental fields. Quite often, similar benefits can be found in other subject areas than medical and dental fields, for example, the industrial side, such as digital storage for industrial spare parts, which reflects heavily to digital storage of dental data.

Since AM is a class of manufacturing processes, it is important to understand what the bases of these processes are, how those differ from each other and to describe how the process works. This review aims to explore which AM processes and materials are utilized in medical and dental applications. It especially focuses on which processes are less studied to determine research gaps. The limitation of this study is that the aim was not to explore all the possible materials used in the applications.

The current review was guided by the following research questions:What are the basic benefits of AM in medical applications?What AM processes based on ISO/ASTM process classification are utilized in medical applications?What are the example materials utilized in founded process and application combinations?Based on the findings, what are the process and application areas that could show future scientific potential?

## 2. Additive Manufacturing Processes 

The ASTM and ISO standardization organization categorizes the AM process into seven different categories: powder bed fusion (PBF), material extrusion (ME), VAT photopolymerization (VP), material jetting (MJ), binder jetting (BJ), sheet lamination (SL) and directed energy deposition (DED) [1]. Each category includes many different vendors, solutions and material options. In this article, ASTM/ISO categories were followed. This was problematic, since the standard terminology is still not utilized in most studies, and often trade names are used for processes. To clarify different processes and principles, Table 2 lists the names of the process classes and a short description, common starting material form, trade names and how well the process is used to manufacture the plastic type of materials, metals or ceramics. Some of the processes for certain materials are in the development and research phase, such as directed energy deposition VAT photopolymerization and material jetting for metals, and some seem not to exist at all, such as sheet lamination of ceramics or directed energy deposition of plastics and ceramics. It is possible that there are scientific studies and trials of these, but no commercial providers exist. Commonly, new process and material combinations are developed based on demand, which highlights large industries and a substantial need. Usually, this leads to the selection of a commonly used material since that can be utilized in many areas.

Each AM process and piece of equipment require a 3D model of the object that they will manufacture, and the most used format for that is stereolithography, standard triangle language, standard tessellation language (STL). The STL model is then sliced into layers and further processed to commands for the specific AM machine. To additively manufacture the part, a raw material is required, such as power, filament, liquid, paste sheet or pellets. The raw material can then be, for example, melted, dispensed, cured or fused to make parts on a layer-by-layer basis. Terminology in AM varies and, as an example, the powder bed fusion process can be called selective laser melting (SLM), selective laser sintering (SLS) or direct metal laser sintering (DMLS). For material extrusion, the most used terms are fused deposition modeling (FDM) or fused filament fabrication (FFF). As a first invented AM process, stereolithography (SLA) has been very commonly used for processes in the VAT photopolymerization class, but digital light projection (DLP) is also used if the light source is a DPL projector. Trade names in material jetting are PolyJet and NanoParticle Jetting. Binder jetting is often called 3D printing (3DP) or ColorJet printing (CJP). Sheet lamination processes are laminated object manufacturing (LOM) and ultrasonic additive manufacturing (UAM). Directed energy deposition processes are laser-engineered net shaping (LENS) and electron beam additive manufacturing. In addition, many others exist on the market.

## 3. Medical Applications of Additive Manufacturing 

Medical applications of additive manufacturing can be classified in several ways [40,41], but this article follows application classes-based classification. AM applications can be classified into the following classes: “models for preoperative planning, education and training”, “inert implants”, “tools, instruments and parts for medical devices”, “medical aids, supportive guides, splints and prostheses” and “biomanufacturing” [42]. For a more general classification, this can be modified so that implants do not need to be inert, and models for preoperative planning, education and training could also include postoperative and operative models using the term “medical models”. Figure 1 shows an example of an application in each category including a (a) preoperative model of a skull and heart, (b) craniomaxillofacial implants, (c) a dental drilling guide, reduction forceps, nasal and throat swabs, (d) personalized and mobilizing external support and (e) a scaffold for zygomatic bone replacement and resorbable orbital implants.

Classification of medical applications of additive manufacturing:Medical models;Implants;Tools, instruments and parts for medical devices;Medical aids, supportive guides, splints and prostheses;Biomanufacturing.

### 3.1. Medical Models

Medical models are based on patient anatomy, and they can be used for pre- and postoperative operative planning and training; training medical students; and informing patients and patients’ families [25,43]. The geometry can be transformed, for example, by taking only interesting sections or scaling it up or down. If models are used for training, such as bone drilling, haptic response might be desirable to be close to the bone. Medical models are widely used in the craniomaxillofacial area, but there are also cases, for example, from different limbs and other bone structures such as the spine and pelvis [25,44]. If these are utilized in the operating theater, it might be recommended that the models be sterilized, but usually, the material option can be quite freely selected which highlights also that these are one of the most common applications. Figure 2 shows a typical process workflow for manufacturing medical models starting from patient anatomy captured via medical imaging, such as computed tomography (CT), magnetic resonance imaging (MRI) or ultrasound, followed by constructing a 3D model geometry for AM using segmentation algorithms [45,46]. After AM, there is often a need for postprocessing such as removing the support structures. 

### 3.2. Implants

Implants are directly or indirectly additively manufactured to replace defective or missing tissue [47,48]. This class also includes dental applications such as crowns and bridges [49]. The material needs to be tissue-compatible and requirements are strict, and approval processes take a long time. Surface properties might affect cell adhesion. Some of the latest studies have explored how to embed materials inside implants, for example, as a type of drug delivery system [50,51]. In personalized implants, AM is a favorable solution, and a typical process requires the capture of a patient’s anatomy similar to medical models. Then, this digital 3D model of the patient anatomy is used as a design reference to enable patient-specific fitting [52,53]. Most typical implants are made from metals using the powder bed fusion process, and this requires different postprocessing steps such as machining the supports, polishing and heat treatments. Before clinical operation, implants need to be sterilized. Figure 3 shows the typical process flow for implants made by additive manufacturing starting from medical imaging and segmentation followed by 3D modeling of the implant proceeding to AM, postprocessing and sterilization.

### 3.3. Tools, Instruments and Parts for Medical Devices

Tools, instruments and parts for medical devices allow or enhance a clinical operation. They might utilize patient-specific dimensions and shapes, for example, in drilling guides [54], and can be invasive and need a sterilization process, since they can be in contact with body fluids, membranes, tissues and organs for a limited time. This class includes surgical instruments and orthodontic appliances [55,56,57]. One of the largest and most successful businesses in this class is using the VAT photopolymerization process to create molds for vacuum forming clear orthodontic aligners [58]. When patient-specific dimensions are utilized, the process is similar to that of implants and preoperative models from medical imaging or 3D scanning. 3D modeling can be conducted by referring to the 3D model of the patient’s anatomy or from scratch if a patient-specific geometry or fitting is not needed. Postprocessing might include support removal, heat treatments, machining and sterilization. Tools, instruments and parts for medical devices are typically made with the process flow shown in Figure 4. For example, the process starts by taking an impression of the patient’s teeth, 3D scanning it, followed by 3D modeling, VAT photopolymerization AM, postprocessing and using the part made as a mold for soft orthodontic aligners.

### 3.4. Medical Aids, Supportive Guides, Splints and Prostheses

In this class, parts made with additive manufacturing are external to the body, and these can be combined with standard appliances to allow customization. Long-term and postoperative supports, motion guides, fixators, external prostheses, prosthesis sockets, personalized splints and orthopedic applications are examples of applications in this class [59,60,61]. The process can start from medical imaging followed by segmentation, 3D scanning or 3D measurements that can provide data directly for use in the 3D modeling phase. Alternative manufacturing methods for additive manufacturing are quite often computer numerical control (CNC) technologies [62]. Parts may require different kinds of postprocessing depending on the application such as support removal, heat treatments and painting or coating. The typical process flow for medical aids, supportive guides, splints and prostheses using AM is presented in Figure 5. The example case is a personalized and mobilizing external support for a pilon fracture, where 3D modeling is based on measuring the patient’s ankle movement and adjusting the additive manufacturing pieces to locate the hinge so that it controls the movement under force close to the free movement of the ankle.

### 3.5. Biomanufacturing

Biomanufacturing is a combination of additive manufacturing and tissue engineering [63]. Materials need to be biologically compatible and often active with the body so many different polymers, ceramics and composite materials are used [64]. Porous structures with cultivation and a 3D matrix can affect cell specialization. The materials can be osteoinductive, osteoconductive or resorbable [65]. Shapes can be personalized to correspond to defects [66]. For personalized shapes, the patient’s geometry needs to be captured using medical imaging or 3D scanning. In the 3D modeling phase, micro- and macrostructures are modeled, and porous structures are often used for attracting cells and cell growth. The process often needs to be sterile or parts made with the ability to be sterilized after printing. Before final application, there might also be the need for cell growth in vitro or in vivo. Figure 6 shows an example of an orbital floor resorbable implant stating patient geometry with CT and segmentation followed by 3D modeling and AM of the implant. After manufacturing, the implant is sterilized. 

## 4. Different AM Processes in Medical Applications

Different AM processes are utilized in the applications described in Section 3. A search for the applications was first performed using ISO/ASTM terminology with a combination of AM medical application terms in a specific class and followed with a search using trade names or other commonly used names for the processes, such as the manufacturer name, for the hard-to-find application areas for certain process. The aim was to find at least some examples for each category and, moreover, determine which areas lack those applications and processes and why. The databases used for the search were Scopus, Web of Science and Google Scholar. When at least three certain application class and processes were found, the search focus was then directed to other processes and applications. More specific search terms are shown in Table 3.

There are previous studies regarding certain processes and/or application areas of medical applications of AM such as powder bed fusion of metal implants [67], additive manufacturing of medical instruments [55], biomaterials in medical additive manufacturing [68] and medical phantoms and regenerated tissue and organ applications with additive manufacturing [69]. Previous studies have not usually classified the AM processes or reviewed only a single process. Some studies focused only on utilized materials and some only on applications without any information about the AM processes or materials. Based on findings from the literature, Table 4 shows the different AM processes and materials used or explored in the medical application classes formed.

## 5. Discussion and Conclusions

### 5.1. Limitations of Previous Reviews

There are many reviews on the medical applications of AM [28,32,40,55,108,109,110,111,112,113,114,115]. Most of them focus on certain application areas, such as surgical instruments [55], orthopedics [32], cardiology [108], or have a material focus in certain area such as metals for prosthodontic applications [115]. Some studies have looked at different AM processes, for example, material extrusion and binder jetting for producing pharmaceuticals [116] or ceramics for dental applications using material extrusion, binder jetting, VAT photopolymerization or powder bed fusion processes [111]. Furthermore, some only looked at the general applications of AM [112] or the whole process chain developments including, for example, the design phase [40]. There are reviews which focus on applications and even categorize those in some of the AM processes, such as VAT photopolymerization, powder bed fusion and material extrusion [109,110], but the other processes are missing. In dentistry, there are reviews with a material focus, categorizing different AM processes, such as material extrusion, binder jetting, VAT photopolymerization, powder bed fusion, material jetting and binder jetting [28,113], or focusing tightly, for example, on metallic implants in dentistry utilizing the powder bed fusion process [114]. 

However, usually, the limitations are that the utilized AM processes are not described at all, trade names, etc., are used to describe the technology or the reviews do not tie the applications to the particular AM processes. This is biggest obstacle for systematic reviews and statistics. As an example, the material extrusion process in the literature can be found with the terms: fused filament fabrication, fused deposition modeling and filament freeform fabrication. These terms are related because Stratasys registered and trademarked FDM; thus, other producers needed to invent an alternative term. Similar challenges also occur in other fields in AM. On the other hand, materials are usually well described, and the most common ones are thermoplastics, photopolymers and metals such as titanium alloys.

### 5.2. Processes Utilized Rarely—DED, SL

Directed energy deposition is utilized mostly only in implants, and even in those, it is quite a rare process. This might be related to the poor accuracy and surface quality. Commonly, the process is used only for metal materials. One possible application would be to explore repairing parts for medical devices, since in other industries, it is already explored for repairing applications [117]. However, strict regulations in medical devices might limit the utilization since each repair might request its own approval process if the repair process is too similar in all cases. The materials usually used in directed energy deposition are metals, such as titanium alloys, for implants and, therefore, it limits its use, for example, in medical models. Sheet lamination is only used for medical models or phantoms [72,95], which makes sense, since it can be used to make full-color realistic looking models [118]. Other areas using starting material as a sheet are challenging, since new material examples are not usually in the form of sheets. On the metal side, the ability to combine different metals in different layers would open new potential, for example, by adding functional possibility through bi-metal parts for certain implants and medical devices [119]. Sheet lamination devices are quite rare, and there are only two different vendors in the market: metal-based Fabrisonic and paper-based Mcor, which seems to be in receivership status.

### 5.3. Well Established Processes—PBF, MEX, VP

Powder bed fusion, material extrusion and VAT photopolymerization are utilized in all categories and are well-established processes for all the categories of medical and dental applications [67,120,121]. This might be explained as they are also the most common processes in the industry; there are affordable devices available for research, and as starting materials to explore, they are easily found in powder, resin, pellet or paste forms. In addition, mixing new materials with existing ones is achievable. For implants, the most common materials are metals, especially titanium alloys and for other applications, polymers such as polyamide, ABS, Nylon and different photocurable resins. Material extrusion of reinforced composites would be interesting material where light weight is required, such as for prostheses, and there are new developments in the material extrusion of metals which could work in many categories [122,123,124,125].

### 5.4. Processes Well Established in Some Application Areas—MJ, BJ

Material jetting is not used for implants and biomanufacturing. The reason for this may be that the technology is quite challenging, as the material needs to be pushed through multiple small nozzles similarly to inkjet 2D printing. Otherwise, material jetting is a quite accurate process, so it is very well suitable for medical models requiring high accuracy. Material jetting is also capable of making parts with multiple materials and colors, which is already utilized in medical models [126].

Binder jetting is not utilized for tools, instruments and parts for the medical devices category. The reason might be that parts made with binder jetting are not usually very strong compared to other methods [107,127]. For medical models, it is an excellent technique since it can use colors and does not need support structures and the process is fast and affordable [128]. The most common materials are the gypsum-based powders ZP150 and ZP151. Moreover, metals and biomaterials are used. Currently, HP and Desktop Metal are developing production systems for metal parts based on binder jetting, and this would allow the utilization of the binder jetting process very well also for tools, instruments and parts for medical devices [129]. 

### 5.5. Future Possibilities

In the future, it is recommended to utilize a standard terminology in the research conducted on medical and dental applications of additive manufacturing as well as industrial side [130]. This would allow a more systematic comparison of the utilization of different AM methods. Current and future developments in the AM processes, devices and materials would allow for increasing applications in the medical and dental area (Figure 7). It is possible that some of the processes, such as sheet lamination, will not be used at all in the future, or it might be that there will be new players in the market. However, sheet lamination of metals provides interesting possibilities for multi-metals. Some processes, such as binder jetting and material jetting, have a high potential to be more widely used in medical and dental applications with the possibility to make metal parts. Directed energy deposition has potential in repairing metallic parts and material extrusion for composite parts. Material extrusion and material jetting will have possibilities in multi-material parts. Research gaps and, thus, future research possibilities were determined by utilizing the development and possibilities of AM technologies (Table 2) and comparing those to findings from literature (Table 4).

Future research possibilities in medical AM with different processes can be seen in the following areas:Directed energy deposition—repairing medical parts especially in tools, instruments and parts for medical devices;Sheet lamination—multi-metal parts in medicine, especially in tools, instruments and parts for medical devices;Material extrusion—composite parts and multi-material, especially in medical aids, supportive guides, splints and prostheses;Material extrusion—metal parts especially in implants and tools, instruments and parts for medical devices;Binder jetting—metal parts especially in implants and tools, instruments and parts for medical devices;Material jetting—multi-material parts, especially in medical models and biomanufacturing.

## Figures and Tables

**Figure 1 materials-14-00191-f001:**
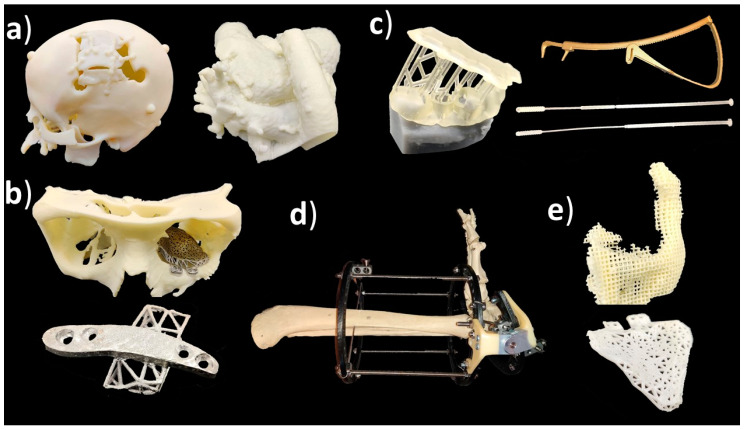
(**a**) Medical models; (**b**) implants; (**c**) tools, instruments and parts for medical devices; (**d**) medical aids, supportive guides, splints and prostheses; (**e**) biomanufacturing.

**Figure 2 materials-14-00191-f002:**
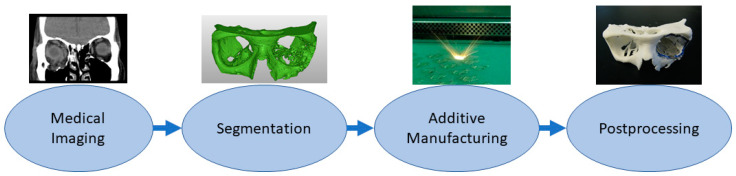
Typical process flow for medical models.

**Figure 3 materials-14-00191-f003:**
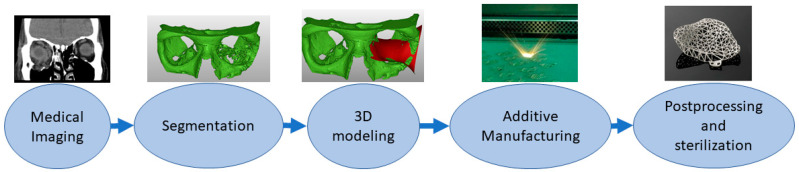
Typical process flow for implants.

**Figure 4 materials-14-00191-f004:**
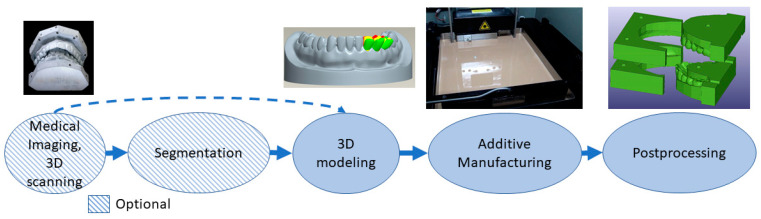
Typical process flow for tools, instruments and parts for medical devices.

**Figure 5 materials-14-00191-f005:**
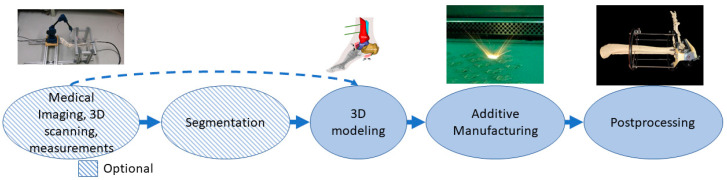
The typical process flow for medical aids, supportive guides, splints and prostheses.

**Figure 6 materials-14-00191-f006:**
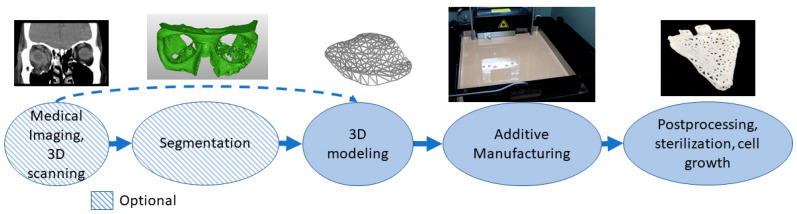
Typical process flow for biomanufacturing.

**Figure 7 materials-14-00191-f007:**
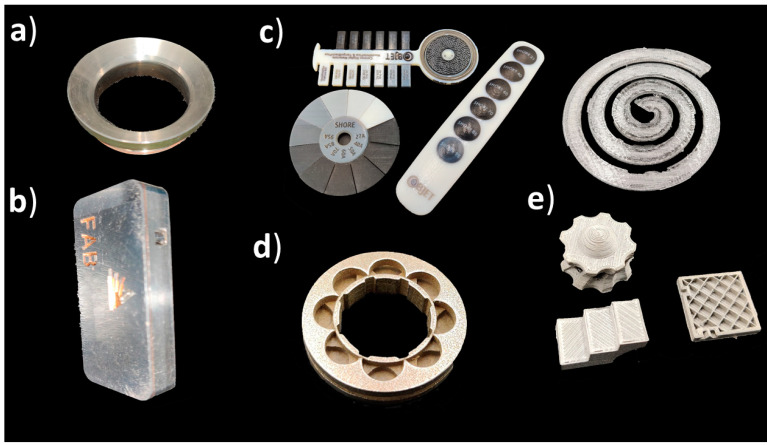
Future possibilities for AM in the medical and dental fields. (**a**) Directed energy deposition repairing, (**b**) multi-material metal parts with sheet lamination, (**c**) multi-material parts with material jetting, (**d**) binder jetting for metal parts, (**e**) material extrusion of metal and composite parts.

**Table 1 materials-14-00191-t001:** Some of the benefits of additive manufacturing (AM) in medical and dental fields.

Reference	Findings	Area
Ballard et al. [22]	cost and time savings	Orthopedic and maxillofacial surgery
Choonora et al. [23]	personalization	Transplants
Mahmoud et al. [24]	cost savings	Pathology specimens for students
Tack et al. [25]	time savings, improved medical outcome, decreased radiation exposure	Surgery
Ballard et al. [26]	incorporation of antibiotics	Implants
Lin et al. [27]	personalization, cost savings	Dental
Javaid et al. [28]	cost and time savings, personalization, digital storage	Dental
Aho et al. [29]	personalization	Pharmacy
Salmi et al. [30]	reduction of manual work	Dental appliances
Aquino et al. [31]	personalization, on-demand manufacturing	Pharmacy
Javaid et al. [32]	accuracy, cost and time savings, personalization, fully automated and digitized manufacturing	Orthopedics
Emelogu et al. [33]	supply chain possibilities	Implants
Gibson et al. [10]	surgeon as designer, innovation potential	Surgery
Haleem et al. [34]	ability to use different materials	Medical
Murr et al. [35]	ability to make complex geometries	Implants
Peltola et al. [36]	template for forming implants	Implants
Ramakrishnaiah et al. [37]	rough and porous surface texture, better stabilization and osseointegration	Dental implants
Nazir et al. [38]	design iterations, supply chain possibilities, complex geometries	Medical devices
Yang et al. [39]	improved understanding of anatomy and accuracy of surgery	Surgery

**Table 2 materials-14-00191-t002:** Characteristics of different AM processes.

AM Process	Short Description	Material Form	Plastics	Metals	Ceramics	Trade/Other Names
Powder bed fusion (PBF)	thermal energy fuses regions of a powder bed	powder	+++	+++	+	selective laser sintering (SLS), direct metal laser sintering (DMLS), selective laser melting (SLM)
Material extrusion (MEX)	material dispensed through a nozzle	filament, pellets, paste	+++	++	++	fused deposition modeling (FDM), (fused filament fabrication) FFF
VAT photo-polymerization (VP)	liquid photopolymer in a vat is cured by light	liquid	+++	+	++	SLA, digital light projection (DLP)
Material jetting (MJ)	droplets of material are selectively deposited	liquid	+++	+	+	PolyJet, NJP
Binder jetting (BJ)	a liquid bonding agent is selectively deposited	powder	+++	++	+	3D printing (3DP), ColorJet printing (CJP)
Sheet lamination (SL)	sheets of material are bonded	sheets	++	++	-	laminated object manufacturing (LOM), ultrasonic additive manufacturing (UAM)
Directed energy deposition (DED)	focused thermal energy used to fuse materials by melting when depositing	powder, wire	-	+++	+	laser-engineered net shaping (LENS), EBAM

Note: +++, widely available/many studies exist; ++, available/several studies exist; +, R&D phase/studies exist; -, no studies exist.

**Table 3 materials-14-00191-t003:** Search terms.

AM Process	Application	and Process Term	or Manufacturer
PBF	medical or dental or implants or surgery or clinical	powder bed fusion or PFB or selective laser sintering or SLS or direct laser sintering or DMLS	-
MEX		material extrusion or fused filament fabrication or FFF or fused deposition modeling of FDM	-
VP		VAT photopolymerization or stereolithography or SLA	-
MJ		material jetting or Polyjet or nano particle jetting	Objet
BJ		binder jetting or Colorjet printing	Zcorp or Zprinter
SL		sheet lamination or LOM or laminated object manufacturing	Mcor or Fabrisonic
DED		directed energy deposition or DED or laser engineered net shaping or LENS	-

**Table 4 materials-14-00191-t004:** Different AM processes in medical applications.

Application Area	PBF	MEX	VP	MJ	BJ	SL	DED
Medical models [44,50,70,71,72,73,74,75,76,77,78,79]	PA, PP	ABS+, PLA	Photocurable resin	VeroWhite, VeroClear, TangoPlus, Multi-material	ZP150, ZP151, PMMA	Paper	
Implants [37,47,49,50,67,80,81,82,83,84,85,86,87,88]	Ti6Al4VTi64, Co–Cr–Mo, Al2O3–ZrO2	PEEK	Clear resin V4, ATZ, NextDent C&B		ZP150, TCP, nickel-based alloy 625, Titanium		Ti6Al4V
Tools, instruments and parts for medical devices [54,55,56,89,90,91,92,93,94,95,96]	PA, Co–Cr, Ti	ABS, ABS+, PLA,	ProtoGen O-XT 18420, Dental SG, Dental LT, Clear resin V2, Photocurable resin WaterShed XC 11122	TangoPlus, HeartPrint Flex, MED610		Paper	
Medical aids, supportive guides, splints and prostheses [60,61,97,98,99,100,101]	PA	ABS, PLA, Nylon	Clear resin, Ciba–Geigy 5170, Somos 6110, Epoxy	Multi-material, Full Cure 720, ABS like, VeroWhite	ZP151, Stainless steel		
Biomanufacturing [63,102,103,104,105,106,107]	PLA, PLGA	PCL, PLA, PLGA, TCP	PDLLA, HA		VisiJet PXL, Calcium phosphate, barium titanate		

Note: polyamide (PA), polypropylene (PP), polyether ether ketone (PEEK), acrylonitrile butadiene styrene (ABS), polylactic acid (PLA), poly lactic-co-glycolic acid (PLGA), polycaprolactone (PCL), tricalcium phosphate (TCP), alumina toughened zirconia (ATZ), poly(DL-lactic acid) (PDLLA), hydroxyapatite (HA), Poly(methyl methacrylate) (PMMA).

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
