# Peer review of "Additive Manufacturing Processes in Medical Applications"

_materials, 2021, doi:10.3390/ma14010191_

Round 1

Reviewer 1 Report

Dear author,

The questions addressed by the research are clearly stated in the manuscript, in agreement with the title: "What are the basic benefits of AM in medical applications? What AM processes based on ISO/ASTM process classification are utilized in medical applications? What are the example materials utilized in founded process and applications combination? Based on findings what are the process and application areas that could show future scientific potential?"

It is reasonably interesting, but not so relevant since the discussion is quite superficial.

It is not a very original work since there are several previous similar reviews. However, it is a complementary study. It is reasonably well-written and generally pleasant to read it, but some sentences could be improved. Hence, a review of the language is recommended. The conclusions agree with the content present in the manuscript and with the addressed questions.

The manuscript presents some relevant information and can be published after minor revision.

Some points to be improved are: 

  1. Even though "materials" is informed in the title, nothing is said about the materials in the abstract.
  2. Review the language. There are some sentences that could be improved (e.g., on lines 33-34).
  3. In Section 2 (p. 88), add the acronyms for the different processes since they are used in Tables 2 and 3.
  4. A clarification of the symbols used in Table 2 is necessary.
  5. Figures 1 to 6 should be further explained in the text.

Reviewer 2 Report

This work is a review on additive Manufacturing processes and materials in medical and dental applications. This work is within the scope of the journal.

Language needs improvement throughout the manuscript. Some of the errors are stated below.

The references in the study are adequate for a review study of this size, but, as stated by the authors, mainly three different topics are reviewed, medical applications, dental applications, and AM materials in medical applications. Since all three topics are very popular with a lot of literature, a review of this size would be more appropriate to focus on just one of the three topics, in order to cover a larger part of the existing literature in each one of the three fields and include also scientific related content for each area, especially since there are several similar reviews in literature.

How this review contributes to the field when compared with several other reviews in the field, should be presented.

The manuscript is interesting to read, but its contents are focusing on a categorization of the presented literature, with the actual scientific merit missing.

Due to the different topics included in the manuscript, the information for each topic is very limited, especially in the materials section and the dental section as well. To this view, the title should be changed and be more focused and additional information should be added throughout the manuscript, or the manuscript should be focused in just one of the three topics mentioned.

Additionally, the following comments need to be considered, prior to the publication of this work:

l21 "For the future, it would recommend" this need to be rephrased

l29 "Additive manufacturing (AM) or more commonly 3D printing" a lot of discussion has been made on whether they are the same, with many claiming they are not, so it is better to rephrase the sentence

l71 "is to explores" this need to be rephrased

l72 "and especially what are not so much used to find research gaps" this is confusing and need to be rephrased

l109 "Usually, each of the AM equipment manufacturers have their own process name, even that the process would belong to one of the 110 process class defined the standard", trivial information for a manuscript to be published in prestigious journal

l159 figure 2, are these images courtesy of the author? If not, license to preproduce them should be acquired, or at least proper citation should be added, unless there are under free license which should be also mentioned. Also, a process flow is presented. This is more or less trivial information, but a reference should be added. Same with the rest of the figures, such as figure 4, etc.

l302 "Fused Filament Fabrication, Fused Deposition Modeling, Filament Freeform Fabrication as a starting." This reviewer agrees with the terminology argument (although terminology is established by itself over time in most cases), but this was an unfortunate example, since this does not refer only to the medical applications at first and mainly because in this case FDM was the term, until Stratays put it under license and afterwards the rest of the companies were obliged to use other terms for this technology. Experts in the field know the exact principle of each technology regardless the term used in each case

l317 "research gaps" how these were determined?

Reviewer 3 Report

This paper makes an extensive review of the state of the art of the literature regarding Additive Manufacturing processes and materials in medical and dental applications. However, it is not clear what is the scientific contribution.

On page 2, the author claims that “The aim of this review is to explorer what materials are not so much used to find research gaps” and “what are the process and applications areas that could show future scientific potential”. However, the section 4 “discussion and conclusions” does not explain the future research prospects. A review paper should identify in a reasoned manner the challenging research areas and research topis. Moreover, this section is too long, it almost looks like an introductory section.

As conclusion, this paper lacks novelty. This investigation is more like a scientific report instead of a research paper, and it is not clear what is the scientific contribution. In consequence, I am afraid that the work does not merit publication in its current form.

Round 2

Reviewer 2 Report

The revised version of the manuscript is significantly improved in its technical aspects. Most of the comments of this reviewer have been adequately replied and corresponding amendments have been made in the revised version of the manuscript. So, manuscript can now be published in its current form.

Reviewer 3 Report

With all due respect, I do not consider the present manuscript suitable for the Materials. Revised version has not provided any significant improvements to subjects novelty. The work provides very limited transferable knowledge.